# Alterations in the Levels of Urinary Exosomal MicroRNA-183-5p and MicroRNA-125a-5p in Individuals with Type 2 Diabetes Mellitus

**DOI:** 10.3390/biomedicines12112608

**Published:** 2024-11-14

**Authors:** Yixuan Fang, Shiyi Sun, Jing Wu, Guanjian Liu, Qinqin Wu, Xingwu Ran

**Affiliations:** 1Innovation Research Center for Diabetic Foot, Diabetic Foot Care Center, Department of Endocrinology & Metabolism, West China Hospital, Sichuan University, Chengdu 610041, China; f2216014698@163.com (Y.F.); sunshiyi2021@163.com (S.S.); wujing9711@163.com (J.W.); 2Chinese Cochrane Centre, West China Hospital, Sichuan University, Chengdu 610041, China; 3Health Management Center, General Practice Center, West China Hospital, Sichuan University, Chengdu 610041, China

**Keywords:** urinary exosomal, type 2 diabetes mellitus, miR-183-5p, miR-125a-5p, biological functions

## Abstract

**Background:** Type 2 diabetes mellitus (T2DM) is a metabolic disorder, and urinary exosomal microRNAs (miRNAs) were utilized as potential disease prediction or diagnostic biomarkers in numerous studies. This study investigated the differential expression of urinary exosomal miRNAs between non-diabetes mellitus (NDM) individuals and those with T2DM. **Aim:** To elucidate the association between urinary exosomal miRNAs and T2DM. **Methods**: We recruited patients diagnosed with T2DM and NDM individuals in West China Hospital, Sichuan University, from November 2023 to February 2024. Subsequently, we performed sequencing of urinary exosomal microRNAs in both groups. The obtained sequencing results were further validated using RT-qPCR in both the training set and the validation set. Additionally, we conducted logistic regression analysis and Spearman correlation analysis on miRNAs with significant differential expression, as well as analysis of their biological functions. **Results**: A total of 118 urine samples were collected, 59 from individuals diagnosed with T2DM and 59 from NDM. There were differentially expressed miR-183-5p (*p* = 0.034) and miR-125a-5p (*p* = 0.008) between the two groups. Furthermore, multivariate regression analysis demonstrated that higher miR-125a-5p levels were negatively associated with the risk of T2DM (*p* = 0.044; OR: 0.046; 95% CI: 0.002, 0.922). Bioinformatics analysis indicated that the target genes of miR-183-5p were predominantly involved in insulin signaling and glucose transport processes, while those target genes of miR-125a-5p primarily mediated autophagy. **Conclusions**: miR-183-5p and miR-125a-5p might be involved in the pathogenesis of T2DM, while higher urinary exosomal miR-125a-5p was negatively associated with the risk of T2DM.

## 1. Introduction

Type 2 diabetes mellitus (T2DM) is a prevalent chronic metabolic disease; according to statistics, in 2017, approximately 462 million individuals were affected by T2DM worldwide, accounting for 6.28% of the global population. It has emerged as the ninth leading cause of mortality and is projected to affect an estimated 783.2 million people by 2045. Furthermore, healthcare expenditures associated with diabetes are anticipated to reach a staggering value of USD 966 billion [1,2]. Additionally, impaired glucose tolerance (IGT) is expected to reach 472 million people by 2030 worldwide [3]. Diabetes with improper glycemic management results in the development of diabetes complications [4]. Early screening, diagnosis, and timely implementation of standardized management were crucial strategies for improving disease prognosis. Currently, the primary clinical tests for diagnosing T2DM include the 75-g oral glucose tolerance test (OGTT) and glycated hemoglobin (HbA1c) [5]. The OGTT primarily assesses fasting plasma glucose (FPG) and 2 h postprandial plasma glucose levels [5]. However, the disadvantage of OGTT was a low response rate [6], and several studies indicated that relying solely on FPG might underestimates the prevalence of diabetes in low- and middle-income countries [7]. Therefore, it was particularly essential to identify more sensitive biomarkers for diabetes and implement proactive measures such as lifestyle interventions.

T2DM is a metabolic disorder characterized by insulin resistance and β cell dysfunction, resulting in dysregulation of blood glucose levels [5]. MicroRNAs, small non-coding RNAs, play pivotal roles in the pathogenesis of various diseases, including diabetes, cancer, and cardiovascular diseases [8,9]. Numerous miRNAs have been demonstrated to be associated with insulin resistance or glucose and lipid metabolism disorders [10,11]. Exosomes serve as intercellular communicators that facilitate the transport of diverse bioactive molecules such as DNA, RNA, proteins, and mRNAs within different body fluids, including urine, blood, etc. [12,13,14]. Urinary exosomal miRNAs were frequently utilized as potential biomarkers for disease prediction and/or diagnosis. For instance, miR-615-3p, miR-145-5p, and miR-27a-3p could be diagnostic markers for diabetic nephropathy (DN), and miR-21 could be a renal fibrosis biomarker [15,16,17]. In the meantime, Min et al. found that miR-29c, miR-146a, and miR-205 had differences between immunoglobulin A nephropathy and healthy controls, which might be regarded as biomarkers for immunoglobulin A nephropathy [18].

In this study, we analyzed the differential expression of urinary exosomal miRNAs in individuals with NDM and T2DM, aiming to evaluate the association between urinary exosomal miRNAs and T2DM. Simultaneously, these findings may provide valuable insights into the underlying pathogenesis of diabetes.

## 2. Materials and Methods

### 2.1. Participants and Sample Collection

A total of 118 participants were recruited for this study consecutively between November 2023 and February 2024, comprising 59 individuals with NDM from the Health Management Center of West China Hospital, Sichuan University, and 59 patients with T2DM from the Department of Endocrinology and Metabolism West China Hospital, Sichuan University. Patients who met all the following criteria were included in the T2DM group: (1) they fulfilled the 1999 WHO diagnostic criteria for diabetes [19]: typical diabetes symptoms with a random plasma glucose ≥ 11.1 mmol/L or FPG ≥ 7.0 mmol/L or 2 h plasma glucose level ≥ 11.1 mmol/L during a 75-g anhydrous glucose tolerance test, (2) relative insulin deficiency, peripheral insulin resistance, and non-essential insulin treatment based on diabetes, and (3) patients who voluntarily participated in this study and signed the informed consent form. However, patients who met one of the following criteria were excluded from the T2DM group: (1) individuals with impaired glucose tolerance or impaired FPG, (2) individuals with type 1 diabetes, gestational diabetes, or other special types of diabetes, and (3) individuals with malignant tumors, undergoing dialysis, or suffering from severe liver, respiratory, or central nervous system diseases. Furthermore, individuals with normal plasma glucose and HbA1c were recruited in non-diabetes group; however, patients with diabetes, malignant tumors, undergoing dialysis, or suffering from severe liver diseases, respiratory diseases, or central nervous system diseases were excluded (Figure 1).

Urinary exosomal miRNAs from participants were tested according to the following procedure (Figure 2).

The morning urine samples (10–20 mL) were collected from all participants and immediately subjected to centrifugation at 2000× *g* for 10 min at 4 °C. Subsequently, the supernatant was stored at −80 °C until use. All procedures followed the recommendations outlined in the position paper by the Urine Task Force of the International Society for Extracellular Vesicles as much as possible [20].

The studies involving human participants were reviewed and approved by the Biomedical Research Ethics Committee of West China Hospital, Sichuan University [No. 2023 (38)]. Written informed consent was obtained from all participants, and the study was registered with the Chinese Clinical Trial Registry (ChiCTR2300077722).

### 2.2. Exosomal RNA Extraction from Urine

Urinary exosomal RNA was extracted by the Urine Exosome RNA Isolation Kit [21] (Norgen, Ontario, Canada, NGB-47200). Briefly, after the urine was removed from the −80 °C refrigerator and dissolved at room temperature, the urine was subjected to two rounds of low-speed centrifugation to remove cell debris and impurities. Subsequently, total RNA was extracted from 10 mL of cell-free urine using spin column chromatography and Norgen’s proprietary resin according to the protocol.

### 2.3. Urinary Exosomal MiRNAs Profile Analyzed by Small RNA-Seq Technology

The SmallRNA-seq technology was performed by Novogene Co., Ltd. (Beijing, China). Agilent 2100 pic600 was adopted to detect RNA’s total amount and fragment distribution. Upon meeting the quality criteria, library construction was carried out using the Small RNA Sample Pre Kit. Briefly, total RNA served as the initial template, and two terminals of small RNA were directly ligated with an adapter based on the particular structure of the 3′ and 5′ terminals, followed by reverse transcription into cDNA. After PCR amplification, the PCR products were subsequently purified with PAGE gel. After the library construction was completed, preliminary quantification was performed using Qubit 2.0 to dilute the library to a concentration of 1 ng/uL. Then, Agilent 2100 was used to detect the insert size of the library, and q-PCR was employed for accurate quantification of the effective concentration (>2 nM) to ensure the quality of the library. Finally, Illumina SE50 sequencing was conducted.

### 2.4. RT-qPCR Analysis

The exosomal RNA was initially subjected to cDNA synthesis using the miRcute Enhanced miRNA cDNA First Strand Synthesis Kit (TIANGEN, Beijing, China, KR211-02). Reverse transcription was performed in a reaction system consisting of 8 μL of total RNA, 10 μL of 2 × miRNA RT Reaction Buffer, and 2 μL of miRNA RT Enzyme Mix, with a favorable procedure at 42 °C for 60 min, 95 °C for 3 min, and then held at 4 °C. The expression levels of miRNAs were quantified using the miRcute Enhanced miRNA Fluorescence Quantitative Detection Kit (SYBR Green) (TIAGEN, Beijing, China, FP411-02). For real-time PCR, a 20 μL reaction mixture was prepared by combining 2 μL of cDNA, 10 μL of 2 × miRcute Plus miRNA PreMix (SYBR&ROX), 0.4 μL of forward primer, 0.4 μL of reverse primer, and 9.2 μL of ddH_2_O, and the reaction mixture was subjected to 1 cycle of 95 °C for 15 min, 5 cycles of 94 °C for 20 s, 65 °C for 30 s, and 72 °C for 34 s, and 45 cycles of 94 °C for 20 s and 60 °C for 34 s min using a CFX96 Touch (Bio-Rad, Hercules, CA, USA). The expression of miR-10b-5p and miR-148a-3p, which showed no significant difference between the T2DM and the control groups based on small RNA-seq results, were used as endogenous controls to normalize the data. Relative expression levels of each target miRNA were calculated using the 2^−ΔCt^ method (2^−(Ct value of target miRNA−average Ct value of miR-10b-5p− and miR-148a-3p)^), and normalized expression values were further log_10_-transformed (log_10_^2^−(Ct value of target miRNA−average Ct value of miR-10b-5p and miR-148a-3p)^) [22].

### 2.5. Blood Index Test and Definition of Other Variables

Blood biochemical parameters were measured using an automatic biochemical analyzer, while blood routine was detected through automated hematology analyzers. HbA1c levels were determined by high-performance liquid chromatography. Current smokers were defined as individuals who had smoked regularly within the past 12 months [23], and current drinkers were defined as participants who had consumed any type of alcoholic beverage within the past 12 months [24].

### 2.6. Gene Ontology (GO) and Kyoto Encyclopedia of Genes and Genomes (KEGG) Pathway Enrichment Analysis

We utilized miRTarBase (https://mirtarbase.cuhk.edu.cn) and miRSystem (http://mirsystem.cgm.ntu.edu.tw) to enrich the target genes of the selected miRNAs. Subsequently, KEGG and GO analyses were performed using Sangerbox 3.0 (http://sangerbox.com/) on 21 October 2024. The chord diagrams depicting significantly enriched GO items and KEGG pathways (*p* < 0.05) were generated using circlize in R 4.3.2 as well as Sangerbox 3.0 (http://sangerbox.com).

### 2.7. Statistical Analysis

The statistical analysis was conducted using IBM SPSS 26.0 software. Continuous variables were described as mean ± standard deviation, while categorical variables were presented as percentages. The independent samples *t*-test was employed for normally distributed data, the rank-sum test for non-normally distributed data, and the chi-square test for categorical data analysis. Three multivariate binary regression models were constructed. Spearman correlation analysis was utilized to explore the correlation between variables of interest. Statistical significance was defined as *p* ≤ 0.05.

## 3. Results

### 3.1. Clinical Characteristics

The sequencing group comprised eight individuals with NDM and nine patients with T2DM, and there were no significant differences in gender, age, body mass index (BMI), and other demographic characteristics. A training set consisting of 25 NDM participants and 25 patients with T2DM was utilized to preliminarily identify miRNAs exhibiting statistically significant differences through RT-qPCR analysis to validate the sequencing results. Notably, patients with T2DM exhibited higher BMI, systolic blood pressure (SBP), diastolic blood pressure (DBP), triglyceride (TG), and white blood cell (WBC) compared to NDM participants. Additionally, they had lower levels of total protein (TP) and high-density lipoprotein cholesterol (HDLC). Finally, a validation set consisting of 59 participants per group was used to further demonstrate the differential expression of miRNAs between the T2DM and NDM groups. Apart from gender, age, current smoking status, BMI, prevalence of hyperlipidemia, aspartate transaminase (AST), estimated glomerular filtration rate (eGFR), uric acid (UA) levels, and platelet count (PLT), there were significant differences in other variables between the two groups in the validation set (Table 1).

### 3.2. Urinary Exosomal MiRNA Profiling

The volcano plot (Figure 3A) and heat map (Figure 3B) revealed markedly different miRNAs of urinary exosomal miRNAs between the T2DM and NDM groups, with 19 miRNAs exhibiting upregulation and 12 miRNAs displaying downregulation. miRNAs were selected if their average read count was > 50 in two groups, |log_2_ (foldchange)| ≥ 1, and *p* ≤ 0.05. Consequently, five miRNAs, namely, miR-125a-5p, miR-122-5p, miR-183-5p, miR-1-3p, and miR-29a-3p, were selected for subsequent validation.

### 3.3. Validation Results of Differentially Expressed MiRNAs by RT-qPCR

We subsequently performed RT-qPCR analysis in the training set to validate the results of small RNA-seq technology. Due to high Cq values (>35) observed for miR-1-3p and miR-122-5p in most individuals, only miR-183-5p, miR-29a-3p, and miR-125a-5p were confirmed in the training set. The expression levels of miR-183-5p (*p* = 0.048) and miR-125a-5p (*p* = 0.024) in the T2DM group were significantly different compared with the control group (Figure 4). Consequently, we further investigated the expression of miR-183-5p and miR-125a-5p in the validation set. Our results revealed that miR-183-5p (*p* = 0.034) and miR-125a-5p (*p* = 0.008) had a significant difference between T2DM and NDM groups, which were consistent with the sequencing results (Figure 4).

### 3.4. Correlations of Candidate Urinary Exosomal MiRNAs with Clinical Parameters

To investigate the association between clinical indicators and urinary exosomal miRNAs, we performed Spearman’s rank analysis. As shown in Table 2, only miR-125a-5p demonstrated a statistically significant weak association with FPG, HDLC, and HbA1c. The statistically significant results are presented in Figure 5.

### 3.5. Binary Logistic Regression Analysis for the Risk Score

We constructed three models to investigate the association between urinary exosomal miRNAs and T2DM (Table 3). In model 1, miR-183-5p (*p* = 0.036) and miR-125a-5p (*p* = 0.007) exhibited significant associations with T2DM. Model 2 indicated that miR-125a-5p (*p* = 0.006) and SBP (*p* = 0.000) were significantly correlated with T2DM. Finally, miR-125a-5p (*p* = 0.044), SBP (*p* = 0.000), ALT (*p* = 0.018), TP (*p* = 0.000) and WBC (*p* = 0.001) exhibited a significant association with T2DM in model 3. The risk of T2DM decreased by 95.4% for increasing 10 times relative expression (2^−(Ct value of miR-125a-5p−average Ct value of miR-10b-5p and miR-148a-3p)^) of miR-125a-5p compared to the baseline value while keeping other factors constant. Moreover, in model 3 of miR-183-5p, SBP (*p* = 0.000), ALT (*p* = 0.012), TP (*p* = 0.001), and WBC (*p* = 0.001) exhibited a significant association with T2DM.

### 3.6. GO Analysis and KEGG Pathway Analysis

To further evaluate the biological function of urinary exosomes miR-183-5p and miR-125a-5p, we performed GO analysis and KEGG pathway analysis. KEGG pathway enrichment analysis manifested the target genes of miR-183-5p (Figure 6A), which were mainly enriched in several signaling pathways, including TGF-beta signaling pathway (count: 2; *p* = 0.02531605), AMPK signaling pathway (count: 2; *p* = 0.03968579), Hippo signaling pathway (Count: 3; *p* = 0.00743057), and Wnt signaling pathway (count: 3; *p* = 0.00825474). In addition, the target genes of miR-125a-5p (Figure 6B) were mainly enriched in AMPK signaling pathway (count: 4; *p* = 0.00393158), HIF-1 signaling pathway (count: 6; *p* = 0.00002395), Jak-STAT signaling pathway (count: 6; *p* = 0.000218386), and PI3K-Akt signaling pathway (count: 8; *p* = 0.000542758).

GO analysis classified these target genes based on biological process aspects (Figure 7A, B). The BP catalog of miR-183-5p was focused on insulin and glucose transport-related biological processes, including response to insulin (count: 3; *p* = 0.01800634) and cellular glucose homeostasis (count: 2; *p* = 0.03413463). Moreover, the target genes of regulation of autophagy (count: 5; *p* = 0.00757238) were predominantly associated with miR-125a-5p.

## 4. Discussion

By 2030, the prevalence of prediabetes is projected to exceed 470 million individuals, with approximately 5–10% of them progressing to diabetes [3]. Although FPG or HbA1c levels served as convenient screening methods for dysglycemia in the general population, it should be noted that elevated FPG or HbA1c levels may already indicate a prediabetic or diabetic status. Consequently, there remains a lack of standardized approaches for identifying high-risk individuals prior to abnormal plasma glucose testing. The dysregulated expression of miRNAs is implicated in various pathophysiological processes associated with diabetes [25]. In recent years, exosomal miRNAs have demonstrated significant potential for predicting diabetes-related diseases and can exert therapeutic effects through targeted delivery. Specifically, five plasma exosomal miRNAs were identified as predictive markers for gestational diabetes [26]. Additionally, human umbilical cord-derived and mesenchymal stem cell-derived exosomes have shown the ability to ameliorate T2DM by improving peripheral insulin resistance and mitigating β cell destruction [27]. In this study, we discovered that urinary exosomal miR-183-5p and miR-125a-5p exhibited differential expression between individuals with NDM and those with T2DM, indicating their potential involvement in the pathogenesis of T2DM. The urinary exosomal miR-125a-5p might potentially serve as novel biomarkers for T2DM.

Based on the small RNA-seq and RT-qPCR results, we concluded that the level of urinary exosomal miR-183-5p was higher, while the level of miR-125a-5p was lower in patients with T2DM compared to non-diabetic individuals. Previous studies established that miR-125a-5p and miR-183-5p were associated with diabetes. Specifically, miR-125a-5p promotes insulin sensitivity and enhances β cell function in the pancreas while also playing a role in regulating related factors in diabetic retinopathy [28,29]. Under the induction of saturated fatty acids, miR-183-5p might impair insulin signaling, potentially contributing to the development of insulin resistance [30]. Meanwhile, miR-183-5p has been demonstrated to promote upregulation of glycolysis and its associated genes, including *GLUT1* [31]. We believe that the weak correlation between miR-125a-5p and FPG/HbA1c may be attributed to a limited sample size or outliers and the actual correlation may be stronger. In addition, this correlation further supported the association between miR-125a-5p and T2DM. In our subsequent multivariate regression analysis, we found that miR-125a-5p was negatively associated with the risk of T2DM, which was consistent with sequencing results. However, there was no significant correlation between miR-183-5p and T2DM in the final adjusted model due to a limited sample size.

The prediction of target genes and the enrichment analysis of GO and KEGG biological functions were further conducted to elucidate the involvement of miR-183-5p and miR-125a-5p in the pathogenesis associated with diabetes. Regarding GO enrichment, we observed a close association between both miRNAs and insulin, glucose metabolism, and other molecular functions. These findings provided further confirmation regarding their potential role in the pathogenesis of diabetes. The results obtained from KEGG analysis showed that the target genes of the miRNAs were enriched in the PI3K-AKT or Hippo signaling pathway. Numerous studies already demonstrated that the PI3K-Akt pathway plays a crucial role in diabetes pathogenesis by regulating glycogen synthesis, gluconeogenesis, and lipid synthesis [32]. Notably, the exosome-like nanoparticles derived from mung bean sprouts demonstrated potential therapeutic effects on the progression of diabetes by modulating the PI3K-Akt signaling pathway [33]. It was well known that the Hippo signaling pathway regulated pancreatic function [34], thus regulating the pathogenesis associated with diabetes. Large tumor suppressor 2, the core kinase in the Hippo pathway, could be a potential therapeutic target for improving the survival and function of β cells in diabetes [35]. Additionally, our enrichment results also confirmed the involvement of several common signaling pathways involved in diabetes, such as Notch, Wnt, HIF-1, and JAK/STAT signaling pathways, which also regulated the inflammatory response [36,37,38,39,40]. Actually, inflammatory reaction may lead to the occurrence of T2DM through insulin resistance and may aggravate in the case of hyperglycemia, thus promoting the long-term complications of diabetes [41]. Furthermore, through separate miRNA GO enrichment analyses, we identified that *STAT3* could regulate autophagy and is a target gene of miR-125a-5p. Previous studies demonstrated that miR-125a-5p targeted *STAT3* to modulate glycolipid metabolism and autophagy [42,43]. Enhancement of autophagy was shown to ameliorate inflammation, and its modulation was proven beneficial for diabetic cardiomyopathy [44]. In summary, urinary exosomal miRNAs might participate in insulin-related pathways and functions, necessitating further comprehensive investigation to elucidate their underlying mechanisms.

The limitations of the present might need to be presented. Firstly, the sample size of the population included in the study was inadequate. Secondly, three-phase process for selecting RNAs may yield biased results due to potential misclassification in the biomarker screening phase and training set. Thirdly, the T2DM group consisted of inpatient diabetic patients from the Department of Endocrinology and Metabolism, West China Hospital, Sichuan University, who were already experiencing a relatively severe diabetic state and were affected by acute/chronic complications or therapeutic schedule. This study might not fully capture the expression value of these miRNAs regarding their association with the severity of diabetes and expression level. Additionally, due to the lack of follow-up of the included population, an accurate description of the predictive value of urinary exosomal miRNAs was not possible. Therefore, it was imperative to conduct a multicenter case–control study with a larger sample size to further validate its significance.

## 5. Conclusions

Overall, the findings derived from urinary exosomal miRNA indicated a negative association between higher levels of urinary exosome miR-125a-5p and the risk of T2DM. However, further investigations are warranted to elucidate its role in the pathogenesis and progression of T2DM. Considering the predictive and diagnostic value of urinary exosomal miRNA in various diseases, our research groups are currently conducting an additional cohort study to evaluate the prognostic significance of urinary exosomal miRNA in patients with diabetic foot ulcers.

## Figures and Tables

**Figure 1 biomedicines-12-02608-f001:**
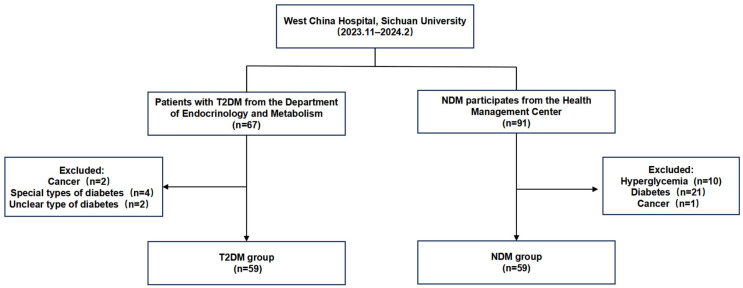
The process of screening participants.

**Figure 2 biomedicines-12-02608-f002:**
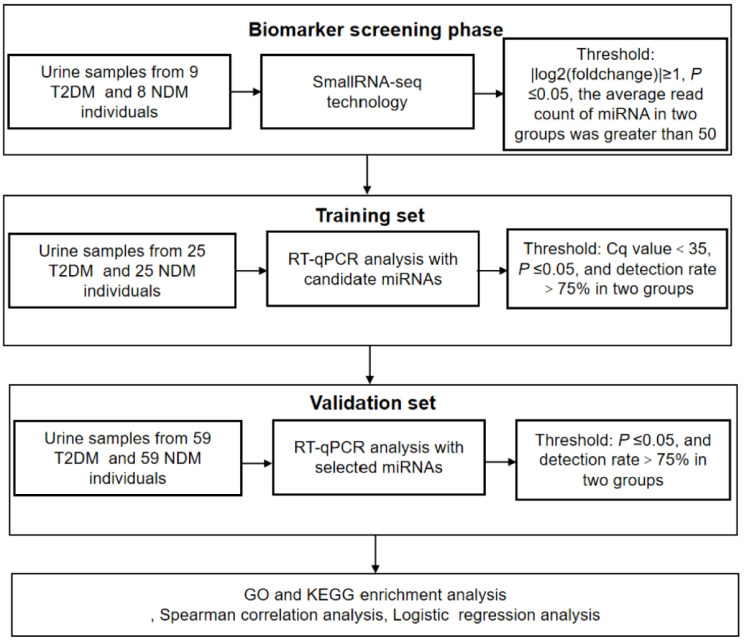
A comprehensive overview of the experimental design.

**Figure 3 biomedicines-12-02608-f003:**
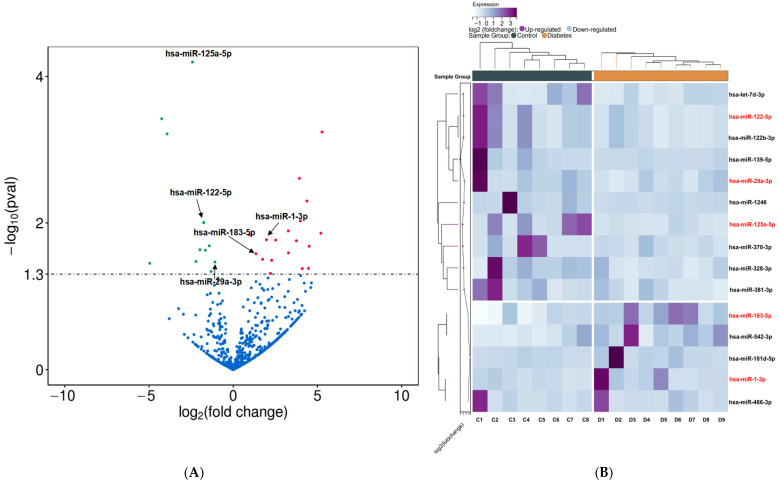
miRNA profiling. (**A**) Volcano plot; green represents decreased miRNA levels, and red represents elevated miRNA levels in the T2DM group; (**B**) heat map.

**Figure 4 biomedicines-12-02608-f004:**
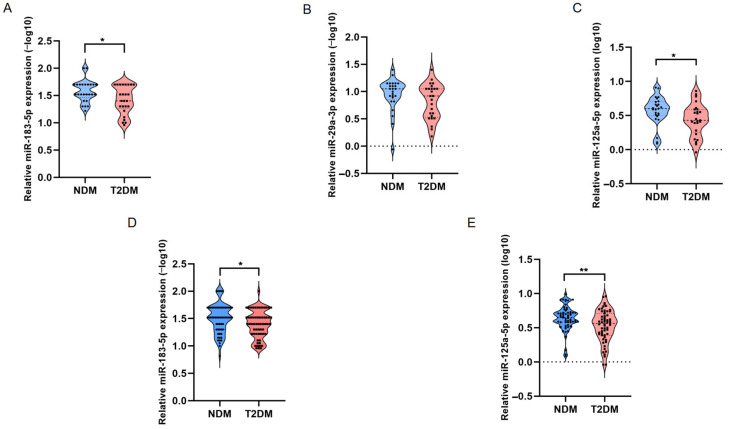
The RT-qPCR results of miRNAs; * *p* ≤ 0.05; ** *p* ≤ 0.01. (**A**) the result of miR-183-5p in training set; (**B**) the result of miR-29a-3p in training set; (**C**) the result of miR-125a-5p in training set; (**D**) the result of miR-183-5p in validation set; (**E**) the result of miR-125a-5p in validation set; NDM: non-diabetes mellitus; T2DM: type 2 diabetes mellitus.

**Figure 5 biomedicines-12-02608-f005:**
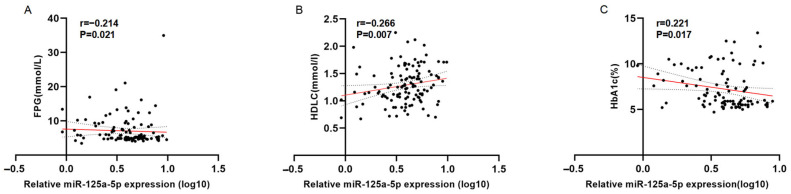
The scatter diagram and fitted curves of Spearman’s rank analysis. (**A**) The result of FPG and miR-125a-5p; (**B**) the result of HDLC and miR-125a-5p; (**C**) the result of HbA1c and miR-125a-5p. FPG, fasting plasma glucose; HDLC, high-density cholesterol; HbA1c, glycated hemoglobin A1c. The red line represents a fitted curve.

**Figure 6 biomedicines-12-02608-f006:**
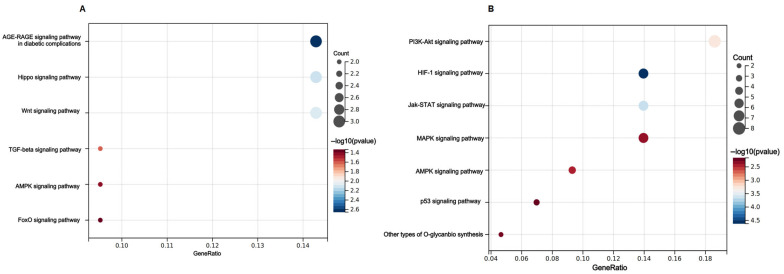
Enrichment results of target genes corresponding to miR-183-5p and miR-125a-5p. (**A**) miR-183-5p KEGG pathway; (**B**) miR-125a-5p KEGG pathway.

**Figure 7 biomedicines-12-02608-f007:**
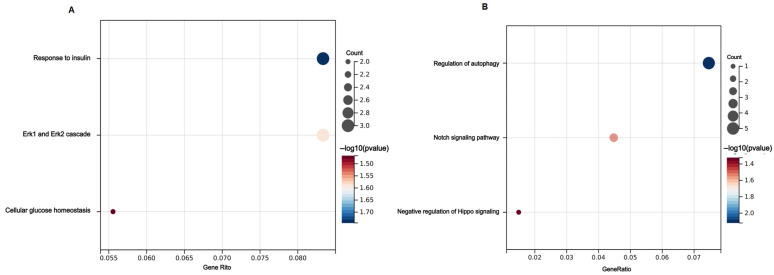
Enrichment results of target genes corresponding to miR-183-5p and miR-125a-5p. (**A**) miR-183-5p GO terms; (**B**) miR-125a-5p GO terms.

**Table 1 biomedicines-12-02608-t001:** Clinical characteristics.

Variables	Sequencing Group			Training Set			Validation Set		
	NDM (*n* = 8)	T2DM (*n* = 9)	*p*-Value	NDM (*n* = 25)	T2DM (*n* = 25)	*p*-Value	NDM (*n* = 59)	T2DM (*n* = 59)	*p*-Value
Gender (Male/female)	5/3	7/2	0.526	13/12	13/12	1.000	34/25	34/25	1.000
Age (years)	60.13 ± 10.05	63.33 ± 10.05	0.521	59.04 ± 6.23	62.40 ± 8.14	0.108	57.81 ± 7.68	60.61 ± 9.95	0.090
Currently smoking	1 (12.5%)	3 (33.3%)	0.576	3 (12%)	7 (28%)	0.157	14 (23.73%)	12 (20.34%)	0.657
Currently drinking	2 (25.0%)	3 (33.3%)	1.000	9 (36%)	6 (24%)	0.355	28 (47.46%)	16 (27.12%)	0.022 *
Duration of diabetes (years)		15.19 ± 8.72			14.92 ± 10.22			13.21 ± 8.99	
BMI (kg/m^2^)	23.59 ± 2.54	22.73 ± 2.30	0.481	23.37 ± 2.97	25.27 ± 3.09	0.031 *	23.57 ± 2.83	24.38 ± 3.07	0.142
SBP (mmHg)	122.50 ± 10.78	141.33 ± 16.19	0.023 *	122.60 ± 8.63	138.76 ± 20.67	0.010 *	122.08 ± 11.83	137.97 ± 20.54	0.000 **
DBP (mmHg)	69.50 ± 9.96	86.33 ± 11.69	0.006 **	74.08 ± 9.27	84.36 ± 10.26	0.030 *	72.09 ± 9.30	83.63 ± 11.55	0.000 **
Hypertension	2 (25.0%)	4 (44.4%)	0.620	4 (16%)	19 (76%)	0.000 **	8 (13.56%)	38 (64.41%)	0.000 **
Hyperlipidemia	2 (25.0%)	4 (44.4%)	0.620	11 (44%)	15 (60%)	0.258	37 (62.71%)	31 (52.54%)	0.264
ALT (U/L)	16.13 ± 4.58	36.56 ± 20.14	0.014 *	19.76 ± 10.28	25.88 ± 16.25	0.059	18.59 ± 8.576	26.37 ± 22.60	0.049 *
AST (U/L)	19.88 ± 3.09	27.00 ± 14.07	0.183	21.28 ± 7.45	20.72 ± 8.98	0.321	19.93 ± 5.620	22.59 ± 14.21	0.957
TB (μmol/L)	14.09 ± 4.82	10.91 ± 3.61	0.152	15.00 ± 4.44	12.60 ± 13.00	0.387	15.02 ± 5.16	12.60 ± 9.20	0.001 **
TBA (μmol/L)	2.93 ± 2.87	4.06 ± 3.55	0.480	3.51 ± 3.48	4.21 ± 2.88	0.139	3.05 ± 2.66	4.16 ± 3.90	0.018 *
TP (g/L)	69.46 ± 4.11	66.80 ± 4.66	0.230	70.82 ± 3.75	66.33 ± 5.81	0.007 **	70.22 ± 3.89	67.13 ± 7.06	0.004 **
eGFR (ml/min/1.73 m^2^)	84.29 ± 19.47	103.25 ± 49.45	0.326	87.47 ± 14.06	85.96 ± 38.68	0.855	87.61 ± 13.45	89.56 ± 33.65	0.681
UA (μmol/L)	336.00 ± 75.82	379.56 ± 151.87	0.475	330.92 ± 81.54	348.84 ± 115.00	0.528	323.60 ± 90.61	350.51 ± 116.08	0.290
TG (mmol/L)	1.28 ± 0.76	3.28 ± 5.71	0.328	1.39 ± 0.70	2.58 ± 3.46	0.010 *	1.54 ± 1.05	2.23 ± 2.65	0.025 *
TC (mmol/L)	4.28 ± 0.94	4.37 ± 1.85	0.900	4.88 ± 1.03	4.65 ± 1.57	0.318	4.99 ± 1.00	4.57 ± 1.43	0.017 *
HDLC (mmol/L)	1.38 ± 0.27	1.13 ± 0.29	0.086	1.45 ± 0.36	1.10 ± 0.31	0.001 **	1.42 ± 0.30	1.15 ± 0.31	0.000 **
LDLC (mmol/L)	2.45 ± 0.91	2.23 ± 0.72	0.599	2.98 ± 0.91	2.64 ± 1.01	0.273	3.11 ± 0.91	2.67 ± 0.98	0.018 *
FPG (mmol/L)	4.77 ± 0.62	8.59 ± 2.48	0.001 **	4.89 ± 0.39	8.78 ± 3.42	0.000 **	4.88 ± 0.37	9.31 ± 5.07	0.000 **
HbA1c (%)	5.55 ± 0.23	8.79 ± 2.82	0.016 *	5.61 ± 0.24	8.53 ± 1.67	0.000 **	5.57 ± 0.29	8.62 ± 2.01	0.000 **
RBC (10^12^/L)	4.70 ± 0.25	4.34 ± 0.47	0.083	4.70 ± 0.47	4.38 ± 0.80	0.096	4.73 ± 0.52	4.37 ± 0.71	0.017 *
WBC (10^9^/L)	5.24 ± 0.91	6.38 ± 1.90	0.157	5.07 ± 1.04	6.94 ± 1.60	0.000 **	5.34 ± 1.27	6.73 ± 1.71	0.000 **
PLT (10^9^/L)	193.25 ± 57.68	215.75 ± 81.14	0.534	208.76 ± 50.16	215.29 ± 67.26	0.703	201.85 ± 48.81	202.98 ± 73.15	0.921
HGB (g/L)	144.75 ± 8.00	132.88 ± 11.74	0.035 *	142.52 ± 9.89	132.42 ± 24.87	0.066	143.81 ± 12.18	132.50 ± 21.81	0.001 *

BMI, body mass index; SBP, systolic pressure; DBP, diastolic blood pressure; ALT, alanine aminotransferase; AST, aspartate transaminase; TB, total bilirubin; TBA, total bile acid; TP, total protein; eGFR, estimated glomerular filtration rate; UA, uric acid; TG, total triglycerides; TC, total cholesterol; HDLC, high-density cholesterol; LDLC, low-density cholesterol; HbA1c, glycated hemoglobin A1c; FPG, fasting plasma glucose; RBC, red blood cell; WBC, white blood cell; PLT, platelet; HGB, hemoglobin concentration; * *p* ≤ 0.05; ** *p* ≤ 0.01.

**Table 2 biomedicines-12-02608-t002:** The result of Spearman’s rank analysis.

	FPG	HbA1c	eGFR	UA	TG	TC	HDLC	LDLC
miR-183-5p	0.035 *p* = 0.704	0.135 *p* = 0.178	−0.099 *p* = 0.299	0.112 *p* = 0.226	−0.029 *p* = 0.753	−0.152 *p* = 0.102	−0.098 *p* = 0.292	−0.180 *p* = 0.053
miR-125a-5p	−0.214 *p* = 0.021 *	−0.266 *p* = 0.007 **	0.161 *p* = 0.088	−0.092 *p* = 0.321	−0.071 *p* = 0.447	0.142 *p* = 0.126	0.221 *p* = 0.017 *	0.156 *p* = 0.092

FPG, fasting plasma glucose; eGFR, estimated glomerular filtration rate; UA, uric acid; TG, total triglycerides; TC, total cholesterol; HDLC, high-density cholesterol; LDLC, low-density cholesterol; * *p* ≤ 0.05, ** *p* ≤ 0.01.

**Table 3 biomedicines-12-02608-t003:** Multivariate logistic analysis of urinary exosomal miRNAs.

Variables	Model 1		Model 2		Model 3	
	OR [95% CI]	*p*-Value	OR [95% CI]	*p*-Value	OR [95% CI]	*p*-Value
miR-183-5p						
miR-183-5p (relative expression(log_10_)) ^1^	4.785 (1.111, 20.611)	0.036 *	4.666 (0.903, 24.108)	0.066	4.201 (0.425, 41.494)	0.219
Male (vs. female) ^2^	-	-	1.052 (0.453, 2.446)	0.906	0.877 (0.270, 2.851)	0.827
Age (years)	-	-	1.005 (0.955, 1.057)	0.858	1.016 (0.937, 1.101)	0.703
BMI (kg/cm^2^)	-	-	1.004 (0.866, 1.163)	0.962	0.885 (0.718, 1.091)	0.254
SBP (mmHg)			1.059 (1.029, 1.091)	0.000 **	1.106 (1.046, 1.169)	0.000 **
ALT(U/L)	-	-	-	-	1.074 (1.016, 1.135)	0.012 *
TB (umol/L)	-	-	-	-	0.939 (0.869, 1.015)	0.115
TP(g/L)					0.813 (0.720, 0.917)	0.001 **
TG (mmol/L)					1.270 (0.849, 1.897)	0.244
LDLC (mmol/L)					0.496 (0.227, 1.082)	0.078
WBC (10^9^/L)	-	-	-	-	2.464 (1.478, 4.108)	0.001 **
miR-125a-5p						
miR-125a-5p (relative expression(log_10_)) ^1^	0.071 (0.010, 0.480)	0.007 **	0.037 (0.004, 0.384)	0.006 **	0.046 (0.002, 0.922)	0.044 *
Male (vs. female) ^2^	-	-	1.152 (0.482, 2.758)	0.750	0.921 (0.277, 3.061)	0.893
Age (years)	-	-	1.003 (0.954, 1.054)	0.914	1.008 (0.932, 1.089)	0.851
BMI (kg/cm^2^)	-	-	0.964 (0.830, 1.121)	0.635	0.843 (0.678, 1.047)	0.122
SBP (mmHg)			1.067 (1.035, 1.101)	0.000 **	1.123 (1.059, 1.191)	0.000 **
ALT(U/L)	-	-	-	-	1.072 (1.012, 1.135)	0.018 *
TB (umol/L)	-	-	-	-	0.943 (0.868, 1.024)	0.163
TP(g/L)	-	-	-	-	0.795 (0.702, 0.901)	0.000 **
TG (mmol/L)					1.190 (0.788, 1.798)	0.409
LDLC (mmol/L)					0.486 (0.215, 1.099)	0.083
WBC (10^9^/L)					2.500 (1.470, 4.252)	0.001 **

^1^ (log_10_^2^−(Ct value of target miRNA−average Ct value of miR-10b-5p and miR-148a-3p)^). ^2^ The reference was a female. Model 1: non-adjusted; model 2: model 1 + gender, age, BMI, SBP; model 3: model 2 + ALT, TB, TP, TG, LDLC, WBC; BMI: body mass index; SBP, systolic pressure; ALT, alanine aminotransferase; TB, total bilirubin; TP, total protein; TG, total triglycerides; LDLC, low-density cholesterol; WBC, white blood cell; OR: odds ratio; CI: confidence interval; * *p* ≤ 0.05, ** *p* ≤ 0.01.

## Data Availability

Date are contained within the article.

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
