# Peer review of "Alterations in the Levels of Urinary Exosomal MicroRNA-183-5p and MicroRNA-125a-5p in Individuals with Type 2 Diabetes Mellitus"

_biomedicines, 2024, doi:10.3390/biomedicines12112608_

Round 1
Reviewer 1 Report
Comments and Suggestions for Authors
The Role of Urinary Exosomal MicroRNAs miR-183-5p and miR-125a-5p in the Pathogenesis and Early Detection of Type 2 Diabetes Mellitus
This paper provides valuable insights into the potential role of urinary exosomal miRNAs in T2DM. However, expanding the study's sample size, exploring mechanisms in greater depth, and addressing limitations around clinical correlations will further enhance the impact and applicability of the findings.
Suggestions for Improvement
- The current study should aim for a larger and more diverse sample size, preferably across multiple centers, to validate the findings more comprehensively.
- The study could benefit from a deeper exploration of the molecular mechanisms by which miR-183-5p and miR-125a-5p affect glucose metabolism and insulin signaling. This would provide a clearer context for their role in T2DM.
- Given that diabetes is a progressive disease, including a longitudinal aspect would allow the authors to assess how miRNA levels change over time and correlate with disease progression.
- The weak correlations between some of the miRNAs and clinical parameters could be acknowledged more explicitly in the discussion, with a focus on how this might impact the clinical application of these biomarkers.
- While the figures provided (volcano plots, heat maps) are informative, the clarity and readability of these visual elements could be improved, particularly for readers less familiar with bioinformatics analysis.
Comments on the Quality of English Language
nil
Reviewer 2 Report
Comments and Suggestions for Authors
Authors used a cross-sectional study examined an association between urinary exosomal microRNAs and type 2 diabetes. However, this article has not fully answered some of the questions due to insufficient description and inadequate statistical analysis.
First, authors do not use follow-up design, which may lead to reverse causation, as they suggest “Additionally, due to the lack of follow-up of the included population, an accurate description of the predictive value of urinary exosomal miRNAs was not possible.” (L320). However, they named the title “The role of …” (L2), and they also suggested “explore their clinical significance” (L17), “developing T2DM” (L27), “increase” (L273), “reduced” (L273), and “developing” (L285). Authors should rewrite the manuscript, carefully.
Second, authors suggest “Patients who met the 1999 WHO diagnostic criteria for diabetes and were further classified as having T2DM based on relative insulin deficiency, peripheral insulin resistance, and non-essential insulin treatment were included in this study.” (L79), but they do not show details of how to diagnoses participants with and without diabetes. Without details of criteria, it is difficult for readers to understand what they did. Moreover, authors suggest “Nevertheless, several studies have indicated that relying solely on FPG may underestimate the prevalence of diabetes in low- and middle-income countries.[4]” (L51) and “Although FPG or HbA1c levels serve as convenient screening methods for diabetes in the general population, it should be noted that elevated FPG or HbA1c levels may already indicate a pre-diabetic or diabetic status. Consequently, there remains a lack of standardized approaches for identifying high-risk individuals prior to abnormal plasma glucose testing.” (L253), but if so, the authors themselves argue that the results of this study were biased, when they used these methods. Authors should explain how to diagnoses participants with and without diabetes, and add limitations, if necessary.
Third, authors suggested “Participants with other types of diabetes, malignant tumors, undergoing dialysis, or suffering from severe liver, respiratory or central nervous system diseases were excluded.” (L81), but they do not show the number of participants before exclusion of these persons. Without details of sample collection, it is difficult for readers to understand what they did. Authors should add flow chart of the number of participants and excluded persons.
Fourth, authors used 3 phases process to select RNAs, but this selection may lead to biased results, because there may be misclassification in biomarker screening phase and training set. Authors should add limitation in discussion section.
Fifth, authors suggest “The sequencing group comprised a total of 8 individuals with NDM and 9 patients with T2DM, and there were no significant differences in terms of gender, age, body mass index (BMI), and other demographic characteristics.” (L159), but there were differences on other biomarkers such as blood pressure, ALT and HGB. Moreover, authors do not adjust for these biomarkers in table 3, which may lead to biased results. Authors should adjust for biomarkers, which were identified in table 1 as potential confounding factors.
Sixth, authors do not explain meaning of odds ratios (e.g., odds ratio per 1 year increment of age) in table 3. Moreover, they do not show the unit for miR-183-5p, miR-125a-5p in tables 3. It is difficult for readers to understand what they did without details of explanation. Furthermore, authors do not explain reference of gender for odds ratios (i.e., male or female) in table 3. Authors should add explanations in table 3.
Seventh, authors suggest “Considering the non-invasive and convenient nature of urine specimen acquisition, urinary exosomal miRNAs could be utilized in conjunction with FPG to detect early-stage diabetes. In this study, we have made a novel discovery that urinary exosomal miR-183-5p and miR-125a-5p exhibit differential expression between individuals with NDM and those with T2DM. These findings suggest their potential involvement in the pathogenesis of T2DM and propose them as promising non-invasive biomarker for predicting T2DM, thereby serving as valuable indicators for early prevention strategies and clinical treatment evaluation.” (L264), but if they aim to suggest so, they should adjust for FPG and HbA1c in table 3. Authors should rewrite the manuscript, carefully.
Finally, authors described some of sentences without citation or justification as follows; “Additionally, there exists a substantial population at risk for developing diabetes; however, many patients either neglect or fail to promptly manage the disease, resulting in the development of various acute and chronic complications associated with diabetes. Early screening, diagnosis and timely implementation of standardized management are crucial strategies for improving disease prognosis. Currently, the primary clinical tests used for diagnosing T2DM include the 75-g oral glucose tolerance test (OGTT) and glycated hemoglobin (HbA1c).” (P42), “However, due to the inconvenience of OGTT, FPG is commonly employed as the primary screening method for diabetes in most populations.” (P50), “T2DM is a metabolic disorder characterized by insulin resistance and β cell dysfunction, leading to dysregulation of blood glucose levels.” (P56), “Urine possesses several advantages, such as a large sample size, convenient sampling, and non-invasive nature, making it an ideal source of disease biomarkers.” (P63), “the 1999 WHO diagnostic criteria for diabetes” (P79), and “Large-tumor suppressor 2, the core kinase in the Hippo pathway, could be a potential therapeutic target for improving the survival and function of β cells in diabetes” (P301), but it is difficult for readers to judge it without references as evidence for each description. Authors should add references for these descriptions.
Reviewer 3 Report
Comments and Suggestions for Authors
Overview
This study explores the relationship between urinary exosomal miRNAs and the pathogenesis of T2DM. The experimental design is rigorous and complete, the statistical methods are appropriate, the data display is complete, the text expression is fluent, and the clinical trial has been registered. We believe that the article is a relatively complete study. But there are some minor issues that need to be addressed.
Details
1. The PCR results of miRNA in Figure 3 should be represented more appropriately using scatter plots or violin plots.
2. The statistically significant results in Table 2 should be presented in both scatter plots and fitted curves.
3. The title and main content of the article are too exaggerated. The article mainly discovers the relationship between only two miRNAs and T2DM, rather than all miRNAs. Therefore, the article should distinguish and explain this, otherwise readers may mistakenly believe it is the entire miRNA cluster.
4. As is well known, miRNA detection is difficult and unstable, so the author should explain the quality control principles for urine samples in the methodology, as well as other measures to improve the stability of miRNA detection.
5. The study selected 118 volunteers from November 2023 to February 2024, is there a selection bias? Similar limitations need to be analyzed by the author in the discussion.
Comments on the Quality of English LanguageThe English could be improved to more clearly express the research.
Reviewer 4 Report
Comments and Suggestions for Authors
Biomedicines3210468 Fang et al
The role of urinary exosomal microRNAs in type 2 diabetes mellitus
Summary
The authors have studied associations between T2DM and urinary exosomal miRNAs in 118 urine samples (59 NDM controls, 59 otherwise uncomplicated T2DM Chinese patients). It concerns a validation phase from a broader study starting from 9 T2DM and 8 NDM individuals for a SmallRNA-seq biomarker screening phase followed by a training set (25 T2DM, 25 NDM individuals) for RT-qPCR analyses for candidate miRNAs in urinary exosomes. After the validation study authors performed GO and KEGG enrichment analyses. Exosomal RNA was extracted from cell-free morning urine samples (Urine Exosome RNA Isolation Kit, Norgen) by spin column chromatography and with Norgen’s proprietary resin protocol. SmallRNA-seq (Novogen Co) and Agilent 2100 pic600 were used for fragment distribution of RNA. PCR products were purified on PAGE gels and quantified using Qubit 2.0. Finaly the Illumina SE50 sequencing was conducted. RT-qPCR analyses were carried out [after using the miRcute Enhanced miRNA cDNA First Strand Synthesis and Fluorescence Quantitative Detection Kits (TIAGEN)]. miR-10b-5p and miR-148-3p were used as endogenous controls for normalization. miRtarBase and miRSystem were consulted for enrichment analyses of target genes for the miRNAs and GO and KEGG analyses were performed by the Sangerbox 3.0.
In the validation study drinking habits, BPs, ALT (p = 0.049), TB, TBA, TP, lipid fractions, RBC, Hb, and WBC count significantly differed between the T2DM and NDM groups. The volcano plot suggested 5 potential candidate miRNAs: miR-125a-5p, miR-122-5p, miR-183-5p, miR-1-3p, and miR-29a-3p. In the RT-qPCR validation study two miRNAs performed very well: miR-183-5p and miR-125a-5p (significantly differently expressed in both the training and validation studies). miR-125a-5p (in contrast to miR-183-5p with no significant association) was moderately associated (r squared <0.1) with FPG, HbA1c, and HDL-C. In binaire logistic regression models adjusted for age, sex, BMI, AST, TG and eGFR miR-183-5p was not associated with T2DM whereas miR-125a-5p was significantly (p = 0.012, OR 0.079 protective) associated with T2DM. KEGG pathway enrichment of miR-125a-5p incriminated genes involved in the Notch, HIF-1, Insulin, Jak-STAT and PI3K-Akt signaling pathways. GO analysis revealed that for miR-125a-5p target genes were mainly involved in the regulation of autophagy and in glucose import. The authors suggested that urinary exosomal miR-125a-5p may potentially serve as a novel protective factor for T2DM.
Comments
1. The authors have studied an interesting and important topic in the search for mechanisms underlying T2DM: miRNAs. It holds true that urinary exosomes are a potentially important source of information for that study question. However it is also a difficult-to-study topic as both the isolation of urinary exosomes and demonstrating a causal involvement of urinary EV-miRNAs in the mechanisms underlying T2DM is a cumbersome “road-to-hell”. The authors incriminated miR-183-5p and particularly miR-125a-5p (protective) as potential key players in mechanisms underlying T2DM or its complications. The authors are not the first to delineate a role for miR-125a-5p in T2DM e.g. ameliorating hepatic glycolipid metabolism and several more. However the role of the 2 incriminated miRNAs through urinary EV-miRNAs has not been extensively studied. In a recent review (Chao et al, J Diab 2023) on EV-miRNAs in T2DM the two miRNAs have not been mentioned as candidates neither for pre-diabetes nor for T2DM. Thus there might be an aspect of novelty (provided the authors’ results hold true). However after reading the manuscript the reviewer has a number of questions and comments.
2. The reviewer is surprised that the authors only retrieved two miRNAs which are closely associated with T2DM. In literature (See also the review by Chao et al) a long list of candidates have already been retrieved and published. The authors retrieved less commonly studied urinary EV-miRNAs but not the well-appreciated urinary EV-miRNAs. Why not? Comment on that?
3. It is well known that isolation of urinary exosomes and isolation of the miRNAs might be a cumbersome procedure. The authors followed a protocol (Norgen) which is only superficially described (lines 94-100). Nonetheless the isolation procedure and extraction are extremely important for the resulting yield of miRNAs. Describe the steps for quality control. Which is the spectrum of the size of the urinary exosomes (after centrifugation and spin column chromatography)? How is the reproducibility of the procedure? Referring to a proprietary protocol does not answer the question. Did the authors follow the recommendations by “Urinary extracellular vesicles: A position paper by the Urine Task Force of the International Society for Extracellular Vesicles” by Extracell Vesicles, 2021)?
4. It is a pity that the authors did not also study a group of patients with pre-diabetes to look at the initiating mechanisms in T2DM.
5. Lines 131-137. HbA1c is a key parameter. Add the methodology used for HbA1c.
6. Fig 3. Although significant differences those differences in relative expression of the two key players between T2DM and NDM are not impressive!
7. The reviewer’s second major concern is the multivariate logistic regression models for miR-183-5p and miR-125a-5p (particularly models 3). The reviewer does not understand at all the choice for the covariates age, sex, BMI, AST, TG and eGFR. Table 1 for the validation group does not suggest at all those covariates: AST (NS), eGFR (NS), BMI (NS), TG (far from being the most significantly different lipid parameter), age (NS), sex (NS). The reviewer expected relevant covariates such as BP, drinking, TB (TBA), TP, HDL-C, Hb, RBC, WBC counts. The reviewer is not at all confident that a more appropriately adjusted model 3 will lead to a significant association between miR-125a-5p and T2DM. Prove it!
8. KEGG pathways. The incriminated signaling pathways for miR-125a-5p (Notch, HIF-1, Jak-STAT, PI3K-Akt) are very non-specific for T2DM and are involved in a long list of (inflammatory) diseases. Thus a word of caution might be needed.
9. Table 2. Associations between miR-125a-5p and poorly regulated common lab parameters in T2DM are not strong at all (R squared < 0.1). Discuss it.
Round 2
Reviewer 2 Report
Comments and Suggestions for Authors
Authors revised the manuscript, but this article has not fully answered some of the questions due to insufficient description and inadequate statistical analysis.
First, authors suggest “due to the fact that the miRNA levels we detected were relative quantitative levels based on endogenous reference (miR-10b-5p and miR-148-3p), whose expression had no difference between T2DM and NDM group, there are no units available.”, but it is difficult for readers to understand the meaning of odds ratios. For example, by using the log-transformation of miR-10b-5p and miR-148-3p as a predictive variable in logistic regression models used for Table 3, you can calculate odds ratios for 10% increment of miR-10b-5p and miR-148-3p. As authors used the log-transformation in Figure 4 and Figure 5, it may be better way to calculate odds ratios.
Second, authors suggest ““Male (vs. female)” Page 9 Line 239”, but they do not add “(” and “)” in Tabel 3. For the better understanding fore the readers, authors should explain that the reference is females in the footnote of Table 3.
Authors should revise the manuscript.
Reviewer 4 Report
Comments and Suggestions for Authors
The authors have addressed all questions and comments which have been raised by the reviewer and adequately revised their manuscript. The manuscript has improved. The reviewer does not have further comments/questions.
Round 3
Reviewer 2 Report
Comments and Suggestions for Authors
Authors revised the manuscript, but this article has not fully answered some of the questions probably due to inadequate statistical analysis.
In fact, authors suggest they used the log-transformation of miR-10b-5p and miR-148-3p as a predictive variable in logistic regression models used for Table 3, but the odds ratios as well as 95 % confidential intervals are the same before and after revise. Authors should check the odds ratios they calculated.
Round 4
Reviewer 2 Report
Comments and Suggestions for Authors
Authors revised the manuscript, and I have no further comment.